# 'In search of lost time': Identifying the causative role of cumulative competition load and competition time-loss in professional tennis using a structural nested mean model

**Stephanie A. Kovalchik** *

Zelus Analytics, Austin, Texas, United States of America

* skovalchik@zelusanalytics.com

**Data Availability Statement:** The data used in this study are available from the OnCourt database. Both one-year and lifetime licenses to access these data can be purchased at http://oncourt.info.

## Abstract

Injury prevention is critical to the achievement of peak performance in elite sport. For professional tennis players, the topic of injury prevention has gained even greater importance in recent years as multiple of the best male players have been sidelined owing to injury. Identifying potential causative factors of injury is essential for the development of effective prevention strategies, yet such research is hampered by incomplete data, the complexity of injury etiology, and observational study biases. The present study attempts to address these challenges by focusing on competition load and time-loss to competition—a completely observable risk factor and outcome—and using a structural nested mean model (SNMM) to identify the potential causal role of cumulative competition load on the risk of time-loss. Using inverse probability of treatment weights to balance exposure histories with respect to player ability, past injury, and consecutive competition weeks at each time point; the SNMM analysis of 389 professional male players and 55,773 weeks of competition found that total load significantly increases the risk of time-loss (HR = 1.05 per 1,000 games of additional load 95% CI 1.01-1.10) and this effect becomes magnified with age. Standard regression showed a protective effect of load, highlighting the value of more robust causal methods in the study of dynamic exposures and injury in sport and the need for further applications of these methods for understanding how time-loss and injuries of elite athletes might be prevented in the future.

## Introduction

Injury is one of the most significant threats to the longevity of elite athletes and, when injury ends the careers of the industry's stars prematurely, can pose a significant threat to the business of sports [1]. Continued growth of the sports market has resulted in increasing commercial opportunities and, inevitably, greater physical demands on athletes to play harder, faster, and longer [2]. The growing pressure to stay competitive has made injury prevention a top priority of high-performance sport [3].

The demands of play have reached a crisis point for professional tennis in recent years. At the end of 2018, a spate of injuries at the top of the men's sport saw multiple of the highest ranked players end their seasons early, several spending more than six months away from the

Licenses are available to any users and any one with a license can access all of the data used in this research.

**Funding:** The author received no specific funding for this work. Tennis Australia provided support in the form of a salary for author SAK, but did not have any additional role in the study design, data collection and analysis, decision to publish, or preparation of the manuscript. The specific roles of this author are articulated in the 'author contributions' section.

**Competing interests:** The author SAK was employed by Tennis Australia during the completion and writing of this study. This does not alter the author's adherence to PLOS ONE policies on sharing data and materials.

sport [4]. It remains unclear whether these prominent cases are indicative of a broader rise in injury risk, yet several characteristics about the structure of the sport make it vulnerable to systemic fluctuations in its risk profile [5]. Firstly, tennis events are spread throughout the globe and top players are often travelling long distances between events. Secondly, matches themselves have no theoretical limit and the actual duration of a match can change dramatically depending on the match format, surface, and other tournament factors. Thirdly, the calendar of professional events is constantly in flux, with most changes resulting in a more congested season where there are fewer opportunities for recovery [6]. The 'grind' [7] that the season schedule imposes is most pronounced for the best players as they are the ones expected to last through the most rounds at each tournament.

Etiological models of injury in sport describe a multifactorial process characterized by multiple intrinsic and extrinsic variables [8]. The complexity of the mechanisms of injury makes it clear that no single causative factor can explain all injuries. However, there is consensus among governing bodies of sport that the volume and intensity of physical activity, referred to as 'load', is a major risk factor [9]. This consensus is equally prevalent in tennis where load management is a central tenet of injury prevention [10]. Despite general agreement about the importance of load, there is little agreement on how load is best distributed over time to protect against serious injury [11].

Research into the causal relationship between load and injury in tennis faces many challenges. The individualized nature of the sport makes consistent collection of injury events rare. And injuries that may be documented by physicians or athletic trainers at specific events do not follow a standard protocol, making the comparability of injury data between events questionable [5]. Gathering high-quality data about load is also a challenge. One reason for this is the lack of agreement on how 'load' is defined. Load can take different meanings depending on the experts who are using it [12]: biomechanists use load to focus on the frequency and force of stress to joints, physiologists use load to refer to the respiratory demands on the cardiovascular system, while sports scientists use load to refer to total accelerations performed. Under any definition, a complete picture of the load an athlete may experience over time is rarely available owing to the difficulties of collecting data during the training periods of top athletes [13, 14].

Even addressing the incompleteness and inconsistency of data on injury and its risk factors would not eliminate the hurdles to studying the causes of injury in tennis. Because of the observational nature of tennis data, any analysis would be vulnerable to multiple biases. Owing to single-elimination tournament designs, for instance, more talented players 'select' into higher levels of load but the same players could have better ways to protect against high load (through better movement or recovery practices, for example) such that naive association studies could find load to be associated with fewer injuries. It is well-established that such scenarios generally do not allow the identification of potential causal effects from observational data with standard techniques, like regression analysis [15]. Proper accounting of observational bias is especially difficult in the load-injury setting because of the complexity of load—an exposure of variable intensity that is constantly changing over time and where future exposure may depend on prior exposure [16]. The analytical challenges inherent in modelling a dynamic, continuous exposure is further compounded by the potential for multiple other time-varying factors—player ability, past injury, age, etc.—to moderate or confound the direct effect of load on future injury risk.

The present paper takes several steps to address the above challenges. First, we focus on a well-defined subset of experienced load and outcomes that are completely observable for top professional players; namely, competition load and competition time-loss. Competition load, a type of external load, measures the volume and intensity of professional play throughout a

player's entire professional career, allowing the examination of both acute and chronic cumulative effects of load. Time-loss, an extended break from competitive play suggestive of an unintended absence, will be the outcome of focus because, poor health is the primary cause of missed competition at the elite level [17, 18]. Moreover, like injury itself, time-loss is an outcome that top tennis players want to minimize.

The second major contribution of this paper is the use of a more principled approach to studying the effects of cumulative load on the risk of time-loss. Specifically, we propose a structural nested mean model (SNMM) to estimate the potential causal effect of load. Like marginal structural models (MSMs), the SNMM when combined with inverse probability of treatment weights, can help to address selection and confounding biases when evaluating the effects of time-varying exposures [19]. SNMM are particularly useful when the primary exposure of interest may be moderated by another time-dependent variable [20]. This is relevant in the present study where age is expected to moderate the risk of a given level of accumulated load. In what follows, we develop an SNMM for the potential causal effect of cumulative load on time-loss risk in the presence of age modification using regression with residuals and an inverse probability of treatment weighting strategy.

## Methods and materials

### Sample

Competitive results of men's professional tennis players from 1990 to the present were obtained from the OnCourt database (www.oncourt.info). These data include unique identifiers for the winners and losers of matches, the date of each competition, and the score, which includes the total games and sets played. Player ratings were derived from the first recorded match results in this database using a previously described Elo-based rating system, a statistical algorithm for rating the latent ability of tennis players that accounts for surface and margin of victory [21, 22].

Because the main risk factor of interest in this study was cumulative professional play, it was important to have complete competitive history for players included in the analysis. Results for the lowest-level of competition where money can still be earned professionally, ITF Futures events, are not represented in the database until 2004, only years from 2005 and onward were considered and only for players whose first professional match was 2005 or later.

An early step in the data preparation was defining a sample of players who are regular competitors. Here 'regular' means, players who, if fit, are expected to play throughout the season. Player schedules are expected to vary considerably with the level of their ability, as this dictates the number of events where they are eligible throughout the year. The regularity in play was explored by grouping player-seasons according to the player's rating at the start of the season, using rating groups of 100 points over the range of 1900 to 3000, forming 11 total groups (Note that the average rating of a professional player is 1500, while players who compete in the main draw of Grand Slams are usually rated 2000 or higher). Given that official rankings (See https://en.wikipedia.org/wiki/ATP_Rankings) are based on a player's best 18 tournament results and that no ATP event outside of the World Tour Finals takes place in the months of November and December, it is reasonable to use 3 weeks as a threshold for the upper bound of between-event gaps of a typical top player. Looking over the seasons of the players in the different ratings groups, it was found that only players rated 2300 or higher in the study's player ratings had the majority of gaps (judged by the 90th percentiles, which include 90% of all observed gap days) less than 3 weeks for at least 9 months of the year (see S1 File).

Given this observation, the sample of players were all of those who attained a player rating of 2300 or higher during the observation period. There were 389 players who met this

criterion. The basic time unit for the sample was the competition week, as most professional events last one week at most. With a traditional survival analysis, comparing the 25% of players with the highest game load against the rest of the player sample would have a power of 80% to detect a 20% risk increase for competition time loss over the base rate of 3%.

## Outcome

The primary outcome of the study was competition time-loss: an extended absence from competition.

Definitions for time-loss from competition were individualized to each player using a linear mixed model of the maximum gap during a calendar month (the largest number of consecutive days a player was not competing in a given month) with player random effects and an unstructured covariance-variance [23]. Since a period between competition could extend over one or more months, the period was assigned to the month when the gap commenced and that month alone. From this regression model, we could obtain the expected value for the maximum number of consecutive days a player spends away from competition in a given month. The model was trained on data for players who had 3 or more seasons at a rating of 2300 or more. For players with fewer than 3 seasons at the minimum rating, the expected maximum gap days was set to the average.

The above regression model provides a player and month specific estimate of the maximum between-event days (here on called 'gap days') under normal conditions. A time-loss event was defined as instances where the actual gap days in a month were 2 weeks longer than expected (4 weeks longer for the month of January, owing to the off-season). This rule was validated against a small sample of former World No. 1 players ($n = 3$) with well-documented injury histories and was found to identify all of their documented absences from competition due to injury. Full details of the outcome determination and validation are provided as S1 File.

Fig 1 shows the 90th percentile range for the gap criteria. The months of March to November show the most consistency, with gaps of 25 to 40 days or fewer being the minimum threshold for an unexpected absence. For February, the range is slightly more, with losses of time greater than 40 days before their first match not being unusual for a subset of players. Inspection of players with longer time-loss in February suggests that this subset are players who did

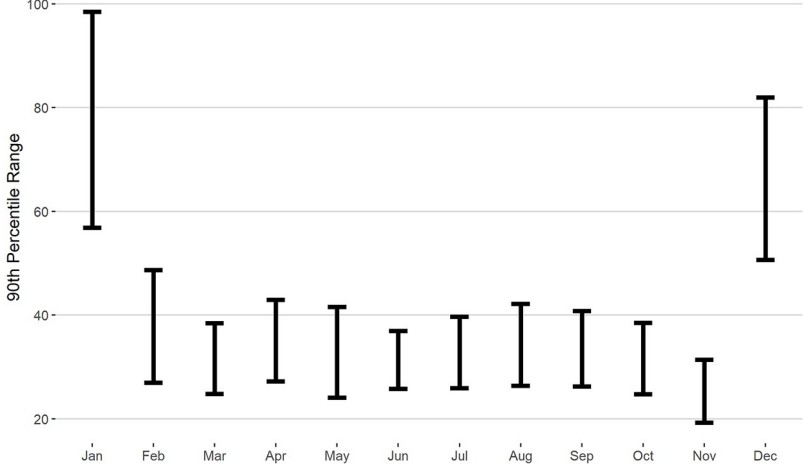

**Fig 1. The 90th percentile range for 'gap days' for each month, which depicts the range containing 90% of the longest periods outside of competition that were observed for each month.**

not tend to compete in the Australian events in January and also players who participated in first round Davis Cup events in early February, a non-tour event that, like exhibition events, is not included in the competition calendar of players in this study. January and December stand out clearly as periods following a player's elected off-season, lasting no more than 60 to 100 days for the typical professional player.

Baseline for monitoring competition time-loss began after a player accumulated 8,000 cumulative games played. Thus all players entered the risk pool with an equal career load.

## Censoring and competing risk

Because a player is only observed when they compete, the last observation is either the last observation in the study observation window, the last observation before an absence in play, or the last professional match.

Retirement, a permanent break from professional play, is a competing risk for temporary absences from competition. Of the 1901 player seasons in the sample, there were 84 instances where a player's last observed professional match was at least one year before the end of the study observation (March 2019). In 36 instances, the final recorded match was in 2018. A review of players whose last observed matches were in 2016 to 2018 revealed 20 players who were not officially retired and 2 who were serving an extended ban. The analysis considered the 20 cases to be instances of an unintended absence, while the remaining 64 cases were treated as censored observations.

## Exposure and moderator

In the absence of confounding, the basic effect modification relationship can be depicted with the directed acyclic graph (DAG) shown in Fig 2 [24]. The target outcome of interest $Y$, which has direct causal effects from $X$. The factor $M$ moderates the process between $X$ and $Y$, as denoted by an edge directed to the edge between $X$ and $Y$. Changes in $X$ are not expected to change the level of the moderator $M$ and $M$ does not have direct influence on $Y$ in the absence of $X$, but the level of $M$ influences the effect of $X$ on the outcome [25, 26].

In the context of competition time lost in professional tennis, the primary causal factor is load (Fig 3). The term 'load' has taken various definitions in the sport injury literature [9]. The present work load will be used to refer to the intensity of competition, a type of external load, which will be measured by the games played during a match.

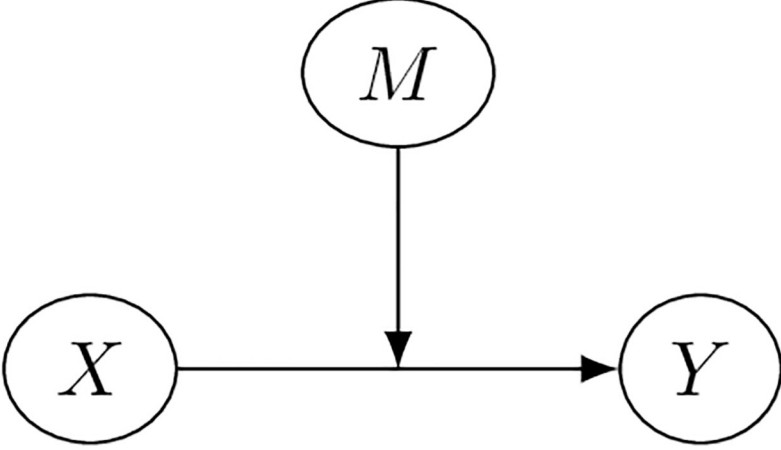

**Fig 2. Point treatment directed acyclic graph for exposure with effect modification.**

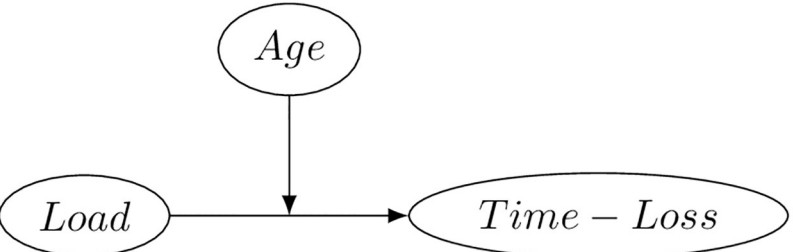

**Fig 3. Point treatment directed acyclic graph for causal effect of load on time-loss with age effect modification.**

The effect of load is presumed to be moderated by an athlete's age, as an equal level of load may become more damaging owing to increase physiological susceptibility associated with age.

A more accurate model for the load-age relationship on competition time-loss accounts for the fact that both factors are changing over time and can be influenced by measured and unmeasured confounders. Let $Y_{\bar{G}(t),\bar{A}(t)}$ be the potential outcome of a time-loss from play given age $\bar{A}(t) = (A_1, ..., A_t)$ and game load history $\bar{G}(t) = (G_1, ..., G_t)$ after the $t$th week of professional competition. The model below is a time-dependent DAG describing the exposure-moderator history in the presence of confounders, assuming that all edges capture the factors that have a causal influence on $Y$.

Fig 4 shows the exposure having direct effects on the target outcome, the moderator having an influence on these effects as well as the values of the primary exposure. The exposure is not

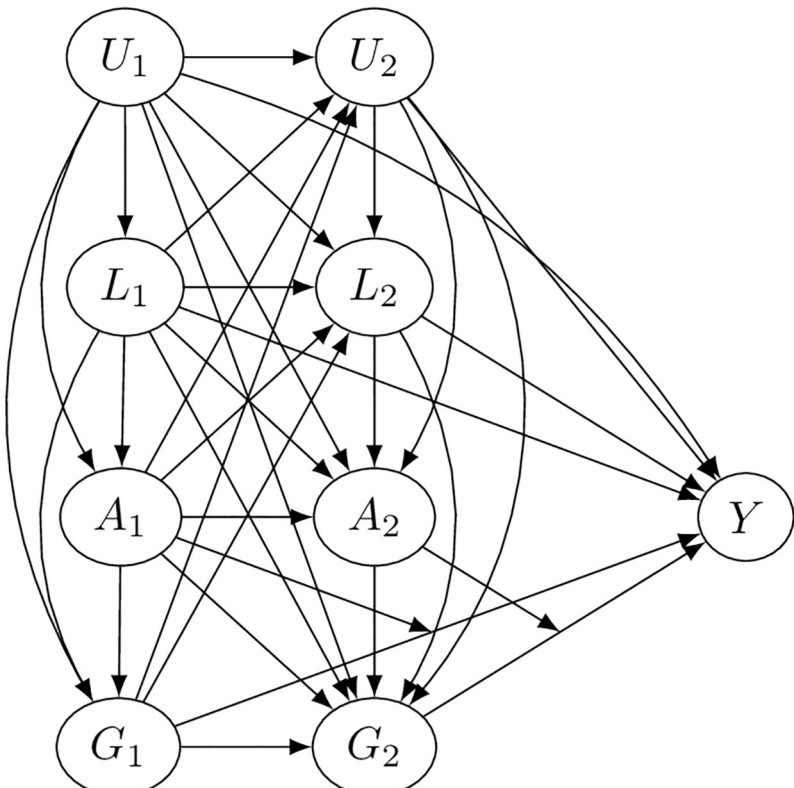

**Fig 4. Directed acyclic graph of presumed causal model for time-varying exposure with effect modification and time-varying measured and unmeasured confounders.**

presumed to have any direct effects on the moderator. However, both the exposure and mediator could be influenced by observed confounders $\bar{L}(t) = (L_1, ..., L_t)$ and unmeasured confounders $\bar{U}(t) = (U_1, ..., U_t)$, which could both have direct effects on the future risk of time-loss from competition.

## Marginal structural model

Our primary interest is in the role that accumulated load has on the risk of time-loss from competition. A player's historical load and age could influence their risk of time-loss in a variety of ways. This paper will consider the role of simple cumulative load. Under this model, the potential outcome of the hazard of a time-loss at time $t$, if load were set to history $\bar{G}(t)$ and age were set to the aging history $\bar{A}(t)$, has the following log-linear dose-response relationship

$$log(\lambda(t|\bar{A}(t), \bar{G}(t))) = log(\lambda_0(t)) + \beta_1 cum(\bar{G}(t)) + \beta_2 A_t + \beta_3 A_t \times cum(\bar{G}(t)) \qquad (1)$$

where $cum(\bar{G}(t)) = \sum_{j \leq t} G_j$ is the total game load through to the $t$th competition week and $A_t$ is the calendar age at the $t$th competition week. There are many possible ways a player could get to the $t$th week with total load $cum(\bar{G}(t))$ and age $A_t$, but the MSM says that all the relevant information for the present hazard is captured by the total load and current age.

Standard MSMs cannot be used to model the effect modification of time-varying covariates [15]. The reason for this is two-fold. First, when a time-varying moderator is correlated with previous levels of treatment, adjusting for the moderator can lead to a biased estimate of the direct and interactive effects of treatment. Second, bias of the direct and interactive effects of treatment could also arise when conditioning on the time-varying moderator owing to unknown causes of the moderator, what is sometimes called a 'collider bias' [27].

An approach for dealing with these sources of potential bias is the structural nested mean model (SNMM) [20]. The SNMM specifies the moderated time-varying causal effects of interest in a conditional mean model for a continuous response given time-varying treatments and candidate moderators. The specific form of the model in the present case can be derived by decomposing Eq (1) into its conditional components. Suppose that the actual level of load received by time $t$ is $g^*$ and the counterfactual age that would be reached by this load is $A_t(g^*)$. The potential hazard can be expressed as,

$$E[log(\lambda(t|A_t(g^*), cum(G\bar{(t)}) = g^*))] = log(\lambda_0(t)) +$$

$$E[log(\lambda(t|A_t(g^*), cum(\bar{G}(t)) = g^*)) - log(\lambda(t|A_t(g^*), cum(\bar{G})_t = g))] + \qquad (2)$$

$$\lambda(t|A_t(g^*), cum(\bar{G}(t)) = g)) - log(\lambda(t|A_t(g), cum(\bar{G}(t)) = g))]$$

a sum of the direct and interactive effects of changing $g^*$ to $g$ when age is fixed and the direct effect of changing $A_t(g^*)$ to $A_t(g)$ when load is fixed. Both of these components are conditional means that, for simplicity, we can assign to the functions $\mu(.)$ and $\epsilon(.)$,

$$E[log(\lambda(t|A_t(g^*), cum(\bar{G}(t)) = g^*))] = log(\lambda_0(t)) + \mu(A_t(g), g^*) + \epsilon(A_t(g^*), g) \qquad (3)$$

The function $\mu(A_t(g), g^*)$ captures the causal effect of changing load at time $t$ for a player of a fixed age. The function $\epsilon(A_t(g^*), g)$ represents the causal and non-causal relationship between the moderator and response, and is considered a 'nuisance function' in the SNMM framework.

In relation to the dose-response model in Eq (1), we model the conditional mean function $\mu$ as a linear function of the cumulative load,

$$\mu(A_t(g), g^*) = \beta_1 g^* + \beta_2 A_t(g)g^* \tag{4}$$

which states that, conditional on a player's age, a unit increase in load has the same change on the log-hazard no matter the time $t$ or the load prior to $t$.

The nuisance function $\epsilon$ has to have the property that $E[\epsilon(A_t(g), g)]$ is zero with respect to the random variable $A_t(g)$. We can guarantee this property by specifying the following residualized form of $\epsilon$. Namely,

$$\epsilon(A_t(g), g) = v_1[A_t(g) - (\alpha_1 + \alpha_2 g)] \tag{5}$$

which says that the conditional mean of age is linearly related to the cumulative load $g$, such that a unit increase in load has the same expected association with age no matter the specific history it took to get to load $g$. This conditional expectation is subtracted from the realization of $A_t(g)$, giving the residual between the observed and expected age, conditional on all prior history of load.

Combining the above, we get the complete log-linear SNMM,

$$E[log(\lambda(t|a, g^*))] = log(\lambda_0(t)) + \beta_1 g^* + \beta_2 ag^* + v_1(a - \alpha_1 - \alpha_2 g^*) \tag{6}$$

Eq (6) looks much like a standard log-linear model with interactions. However, the SNMM is based on a conditional mean of the supposed moderator. It also lacks direct adjustment for time-dependent confounders. Imbalance in these factors are instead handled through the use of inverse probability of treatment weights. The 'treatment', a general term the causal inference literature uses to refer to the main explanatory variable of interest, in this case is the cumulative competition load. Weighting observations by the inverse probability of the observed dose of treatment received is well known to be a more effective strategy for protecting against confounder bias than regression [28].

## Estimation

The estimation of the SNMM begins by preparing the outcomes and covariates of the observed data. Baseline for the player sample began when all players reached 8,000 cumulative games. From baseline, every competition week was collected and summarized until a player's last professional event or the end of data collection (March 2019). For each week, we computed the total game load, player age in years, player rating, whether the player competed in the previous week, and whether they had a time-loss of more than 180 days at any time in their past 30 events played.

The outcome of time-loss was determined at the end of each competition week according to the player-specific criteria described above.

The first step of the SNMM estimation is the derivation of inverse-probability of treatment weights (IPTW). The purpose of these weights is to create a pseudo population that is balanced with respect to confounding variables, like player ability or competitive play, for all time $t$.

Let $g_{it}$ be the observed game load for the $i$th subject in the $t$th competition week. In the absence of censoring, the stabilized weights are given by,

$$sw_{it} = \prod_{j=1}^{t} \frac{f(G_{ij} = g_{ij}|\bar{A}_{i(j-1)}, \bar{G}_{i(j-1)})}{f(G_{ij} = g_{ij}|\bar{G}_{i(j-1)}, \bar{L}_{ij})} \tag{7}$$

Here, $f(.)$ is a likelihood function for some parametric family appropriate for a continuous treatment variable. The denominator's role is to identify individuals who would be unlikely to have received a given level of load, $g_{ij}$, given their covariate history, and to upweight them accordingly. This is how the weights function to balance the sample with respect to time-varying confounders. The numerator does not include any confounding factors and its purpose is solely to provide stability to the overall weights by providing a number on the scale of the denominator.

Let $C(t)$ be the indicator of a player who is censored in the $t$th competition, because of administrative censoring or retirement. Censoring weights take a similar form as in Eq 8, with

$$cw_{it} = \prod_{j=1}^{t} \frac{Prob(C_{ij} = 0|\bar{A}_{i(j-1)}, \bar{G}_{i(j-1)})}{Prob(C_{ij} = 0|\bar{G}_{i(j-1)}, \bar{L}_{ij})} \tag{8}$$

the probability of having made it to the $t$th competition week without having been censored given load and moderator history, in the numerator, and additional confounder history, in the denominator. The final weight assigned at time $t$ is $w_{it} = sw_{it} \times cw_{it}$.

The treatment denominator weights were obtained from the following linear regression model,

$$E(G_{it} = g_{it}|\bar{G}_{i(t-1)}, \bar{L}_{it}) = \delta_0 + \delta_1 g_{i(t-1)} + \delta_2 x_{1,it} + \delta_3 x_{2,it} + s(x_{3,it}, \theta_1) \tag{9}$$

where $x_1$ is the indicator of back-to-back competition weeks, $x_2$ is the indicator of a 180 day time loss in the past 30 competition weeks, and $x_3$ is a player's rating. The function $s(.)$ is a smoothing cubic spline.

The treatment numerator weights were similarly obtained using,

$$E(G_{it} = g_{it}|\bar{G}_{i(t-1)}, A_{i(t-1)}) = \delta'_0 + \delta'_1 g_{i(t-1)} + \delta'_2 a_{1,i(t-1)}. \tag{10}$$

Estimates for the expected means given in Eqs (9) and (10) were fitted using a GEE model with player as clusters and independent covariance structure between players. Let $\hat{m}_{it} = \hat{E}(G_{it} = g_{it}|\bar{G}_{i(t-1)}, \bar{L}_{i(t-1)})$. The denominator is then calculated as $\phi((g_{it} - \hat{m}_{it})/\hat{\sigma})$, where $\phi(.)$ is the density function for the standard normal and $\hat{\sigma}$ is the residual standard deviation of the fitted model.

Numerator estimates are obtained with the same methodology but without the conditioning on time-dependent covariates. Given numerator expectation, $\hat{\mu}_{0it} = \hat{E}(G_{it} = g_{it}|\bar{G}_{i(t-1)}, A_{i(t-1)})$, with dispersion $\hat{\sigma}_0$, the stable weight is calculated as,

$$\hat{sw}_{it} = \frac{\phi((g_{it} - \hat{m}_{0it})/\hat{\sigma}_0)}{\phi((g_{it} - \hat{m}_{it})/\hat{\sigma})} \tag{11}$$

The identical right-hand side models in Eqs (9) and (10) were used for the models of the censoring weights. However, these were fit in a logistic regression with the outcome being the binary indicator of censoring at the end of the $t$th competition.

The stability of the weights $\hat{w}_{it}$ were evaluated graphically by plotting box-plots against the weeks from baseline. Balance was evaluated by calculating the population standard bias (PSB) at each time $t$ for all time-varying covariates. $PSB \leq 0.25$ was set as the criteria for good balance.

A weighted generalized linear model was used to fit the conditional mean model of the moderator at time $t$ given weights $\hat{w}_{it}$. The age residuals were obtained and served as covariates for the model to estimate the SNMM outcome model. For this model, a weighted pooled logistic regression was used [16]. Effects of load and its moderation by age were illustrated by comparing the estimated hazard ratio at the lower and upper 25th percentiles of cumulative load

observed in the sample for players of age 25, 27 and 29. The upper 25th percentile was approximately 3,000 games more for each of these age groups and this was the increased in load used for these comparisons. For all hazard ratios shown, the reference player was a 25 year-old with a total competition load of 10,000 games. Because the conventional standard errors of the logistic model fail to account for the estimation of the residuals and stabilized weights, bootstrap standard errors and confidence intervals were obtained by repeating the weight, residual, and outcome model estimation for 1,000 bootstrap resamples.

SNMM estimates were compared to the standard association models using an unweighted pooled logistic regression of load and age effect modification with and without adjustment for other measured time-varying covariates in this study. An additional analysis, included the same time-varying covariates in the SNMM outcome model, a so-called 'doubly robust estimate' of the potential causal effect of load [29].

Summaries of the absolute risk from the SNMM model were also estimated for a range of ages and game loads, using the baseline rate for players of 25 years and a 10,000 game load. To understand the potential reduction in absolute risk for top players with some of the highest cumulative loads on tour, the absolute risk reduction for a decrease of 1,000 and 2,000 games played from actual games played was calculated for several prominent players.

All data analysis and modeling was performed in the R statistical programming language [30].

## Identifiability

The ability to identify the causal effects of load using the framework presented in this study rests on several assumptions. The first concerns the correct specification of the dose-response model. A central assumption to the simple cumulative load model is the premise that load and aging history is independent of the actual sequence load was received with age conditional on the current total load and age. In mathematical terms,

$$\lambda(t|\bar{G}_t, \bar{A}_t) = \lambda(t|cum(\bar{G}_t), A_t)$$

Related to the above, is the consistency assumption [31]. This states that a player who has the same treatment and aging history must have the same potential outcome,

$$\lambda(t|\bar{G}_t, \bar{A}_t) = \lambda_{\bar{G}_t\bar{A}_t} \; \forall \; \bar{G}_t, \bar{A}_t$$

The next assumption concerns the positivity of treatment. In the case of the cumulative load, it requires that

$$0 < f(\bar{G}_t = g|\bar{G}_{t-1}, A_{t-1}, \bar{L}_{t-1}) < 1 \; \forall \; t, g$$

all possible treatment levels can be observed for any at-risk person at a given time over all time points.

Finally, in the setting of time-varying treatments with observational data, we have to have sequential ignorability of treatment to be able to identify causal effects [19]. This states that,

$$\bar{G}_t \perp \{\lambda(t|\bar{G}_t)|\bar{G}_{(t-1)}, \bar{A}_{t-1}, \bar{L}_{t-1}\} \; \forall \; \bar{G}_t$$

the particular treatment received at time $t$ must be ignorbale given the measured confounders, pre-treatment moderator history, and pre-treatment treatment history. Because sequential ignorability can be violated if any unmeasured confounders are present, it is an assumption that is not verifiable.

## Results

The data sample included 55,773 competition weeks for 389 professional male tennis players (Table 1). Three percent of the weeks in the study sample were followed by a time-loss. Cumulative game loads had an average of 13,615 games and a maximum of 39,303 across the sampled competition weeks. By design, players had a mean rating over 2300 and the majority of players were rated between 2100 and 2500 in any given week. One of every three competitive events were played on the back of another competition, indicative of the congestion of the tennis calendar. Only a small fraction of players were competing when having had a more than 6 month break from competition in the past 30 events played.

Stratifying the sample by the quintiles of cumulative load shows how the time-dependent covariates vary with increasing games played. As expected, age increases steadily with increasing load with most players being age 29 years or older by the time they have accumulated 15,000 games played (Table 2). Positive trends with load were also observed for the 30-event 180 day or more time-loss, which had its highest observed rates of 5% for the two highest quintiles of load, and player rating, which had a greater average with each higher strata of load. No pronounced trend with back-to-back competition was observed.

The combined stabilized weights show good stability over the competition weeks (Fig 5). On the log-scale, the average weight across the time periods was 1.07 and the interquartile-range was an average of 0.5, a moderate amount of variation over the competition weeks. More instability was observable in the latest weeks, where the sample size was smallest. By the 350th competition week, the average weight dropped to 0.2.

In terms of balance, the player rating showed the greatest potential for confounding bias (Fig 6). The unweighted PSB for the player rating had an average of 0.33, a maximum of 1.15 and exceeded the threshold of 0.25 in 63% of the competition weeks. With weighting, the PSB for the rating reduced to an average of 0.08 and never exceeded 1 in any competition week. For 20% of weeks, the PSB was greater than 0.25 even with weighting. However, this rate was just 4% for the first 200 competition weeks. All other covariates were well-balanced across all competition weeks.

Unweighted regression analysis without adjustment for time-varying covariates showed a protective effect with increasing game load (Table 3). Counterintuitively, the estimates from this regression suggested that an increase of 1,000 accumulated games was associated with a *decrease* of 3% risk in time-loss for a player of the same age. Adjusting for time-dependent covariates removed any statistically significant association, supporting the conclusion that, for players of the same age and skill level, game load has no causal influence on risk of time-loss.

**Table 1. Summary of overall sample characteristics.** Unless otherwise noted, the summary is the mean (standard deviation) for all competition weeks.

| Characteristic | Value |
| --- | --- |
| Competition Weeks, n | 55,773 |
| Players, n | 389 |
| Time-Loss % | 3 (17) |
| Game load | 13,615 (4,208) |
| Age | 28 (3.24) |
| Competed in previous week % | 33 (47) |
| Player rating | 2325 (199) |
| 30-event $\geq$180 day time-loss % | 3 (17) |

**Table 2. Summary of outcome and covariates (mean, SD) by quintiles of cumulative game load.**

| Load Quintile | Load | Time-Loss | Age | Competed in Previous Week | Elo Rating | 30-event Time-Loss |
|---|---|---|---|---|---|---|
| Q1 | 8,851 (499) | 3 (16) | 24.8 (1.77) | 34 (47) | 2232 (176) | 0 (0) |
| Q2 | 10,673 (558) | 2 (16) | 26.1 (1.85) | 34 (47) | 2266 (173) | 2 (15) |
| Q3 | 12,821 (689) | 2 (15) | 27.5 (1.96) | 34 (47) | 2318 (175) | 3 (16) |
| Q4 | 15,526 (889) | 3 (17) | 29.3 (2.12) | 33 (47) | 2358 (180) | 5 (23) |
| Q5 | 20,205 (2,767) | 4 (19) | 32.0 (2.39) | 31 (46) | 2452 (213) | 5 (22) |

In contrast to the standard regression analysis, the SNMM found a significant positive relationship between increased load and risk of time-loss (Table 3). The direct effect of an increase of 1,000 games, for example, is estimated to increase the risk of time-loss by 4% (95% CI 0-9%) in the unadjusted and 5% (95% CI 1-10%) for the doubly-robust analysis. Comparisons between the lowest 25th and highest 25th percentiles of empirical load observed at ages 25, 27 and 29 showed even starker effects. Based on the doubly-robust estimates, the hazard ratios of players in the top 25% of load were consistently greater than those of players of the same age but with the lowest 25% of experienced load. At age 25, the hazard ratio of 1.17 (95% CI 1.02-1.35) corresponds to a risk increase of 17%; at age 27, the hazard ratios of 1.28 (95% CI 1.12-1.47) and 1.06 (95% CI 1.03-1.10) correspond to an increase of 21%; and at age 29, the hazard ratios of 1.65 (95% CI 1.38-1.95) and 1.33 (95% CI 1.20-1.47) correspond to an increase of 24%. Though each of these comparisons correspond to a fixed increase of 3,000 games, we see that the risk associated with that same change in load is increasing, indicating the positive effect modification due to age.

Confidence bands for the effect of load over a broader age range show the results of the SNMM in greater detail (Fig 7). The grey regions highlight the 90th interquartile range of observed load for the specific age group in each panel. In the observed load range for ages 25 to 28 years old, the absolute risk ranged from 2 to 4%. For ages 29 to 32, the absolute risk increased to 6 to 8% for players in the highest load range. Over the age of 32, even players in the lowest observed range had an absolute risk over 4%. Among the highest loads, the risk of time-loss was as high as 18% by age 35.

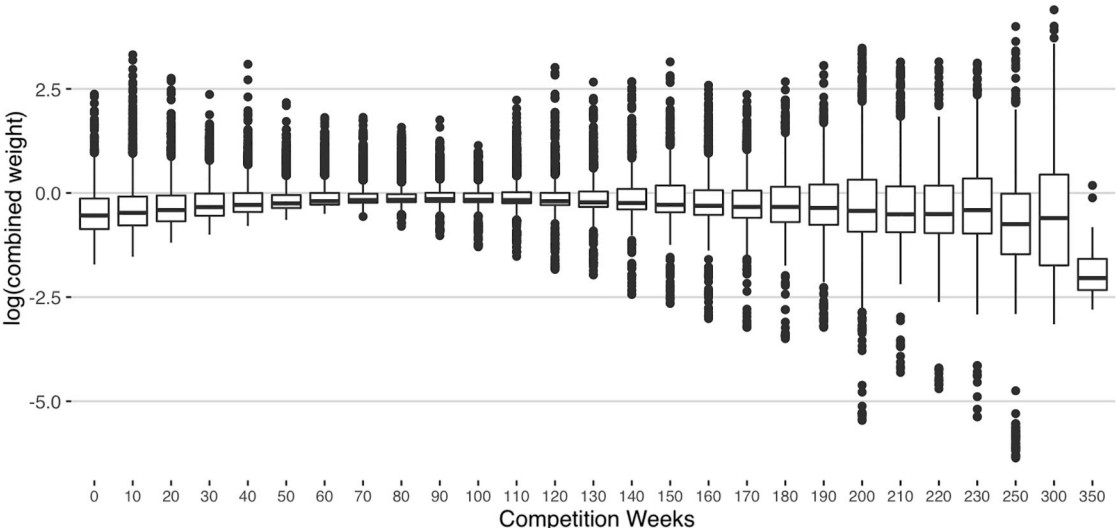

**Fig 5. Log-scale of combined IPTW and censor weights by competition week.**

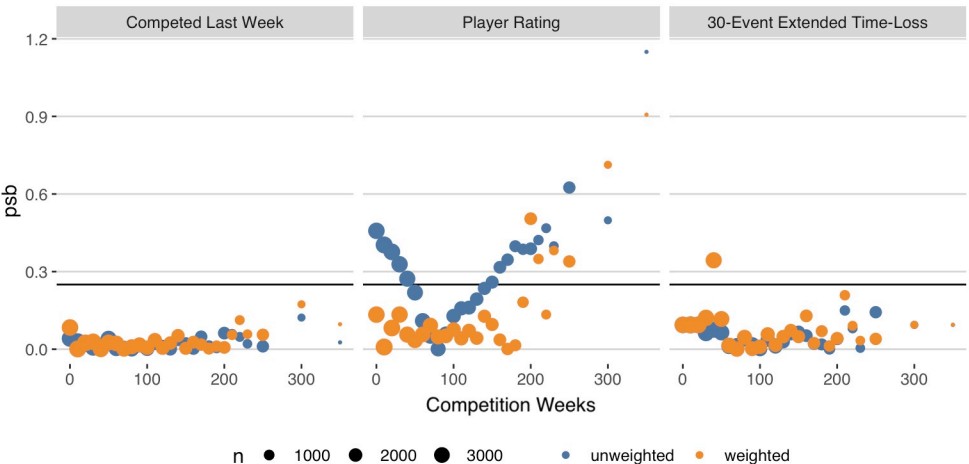

**Fig 6. Population standard bias for time-varying covariates by competition week with and without weighting.** The size of points is scaled to reflect the number of observations at each time point.

Since 2016, several of the most successful male players in tennis have simultaneously suffered long breaks from their usual competition schedule owing to injury. Andy Murray is one among these and is especially notable for having announced his intended retirement at the beginning of 2019. The nature of the sport means that players who consistently perform well tend to accumulate greater load at a similar age. In Fig 8 we see the estimated risk of time-loss for three of the most successful contemporaries—Rafael Nadal, Novak Djokovic, and Andy Murray—given their observed cumulative load at ages 23 to 30 years old. By age 30, Murray had accumulated 24,000 games, Nadal 26,000 games, and Djokovic 27,000 games; levels of load that put them each in the top 10% of load acquired by age 30. At those levels, each of these players would have an expected risk of time-loss for 1 in every 25 weeks of competition. Reductions of 1,000 games would be expected to provide a negligible change in that risk, while 5,000 games fewer of load would be expected to have a more appreciable reduction in the risk of time loss, decreasing the risk to 1 in every 30 weeks of competition.

## Discussion

Despite general agreement about the etiological importance of load for injuries in sport, few studies have examined the longitudinal effects of load [9]. To our knowledge, the present study is the first to investigate the risk of competition load for professional tennis players [32]. Our

**Table 3. Hazard ratio (95% CI) change with load and the effect modification of age for the observed lower and upper 25th percentiles of load for each age.**

| Age | Game Load | Unweighted Unadjusted | Unweighted Adjusted[a] | SNMM Unadjusted | SNMM Adjusted[a] |
|---|---|---|---|---|---|
| Same Age | 1,000 more | 0.97 (0.94-0.99) | 1.00 (0.97-1.03) | 1.04 (1.00-1.09) | 1.05 (1.01-1.10) |
| 25 | 10,000 | 1.00 (Ref.) | 1.00 (Ref.) | 1.00 (Ref.) | 1.00 (Ref.) |
| | 13,000 | 0.90 (0.83 -0.97) | 0.99 (0.90 -1.08) | 1.14 (1.00-1.30) | 1.17 (1.02-1.35) |
| 27 | 11,000 | 0.97 (0.94 -1.00) | 1.00 (0.96-1.03) | 1.06 (1.02-1.09) | 1.06 (1.03-1.10) |
| | 14,000 | 0.89 (0.79-1.01) | 1.00 (0.86-1.15) | 1.24 (1.10-1.43) | 1.28 (1.12-1.47) |
| 29 | 14,000 | 0.92 (0.79 -1.07) | 1.01 (0.86 -1.19) | 1.31 (1.20-1.43) | 1.33 (1.20-1.47) |
| | 17,000 | 0.87 (0.66-1.12) | 1.02 (0.76-1.36) | 1.61 (1.37-1.88) | 1.65 (1.38-1.95) |

[a], Adjusted for player rating, back-to-back competition, and 30-event 180-day time-loss or longer

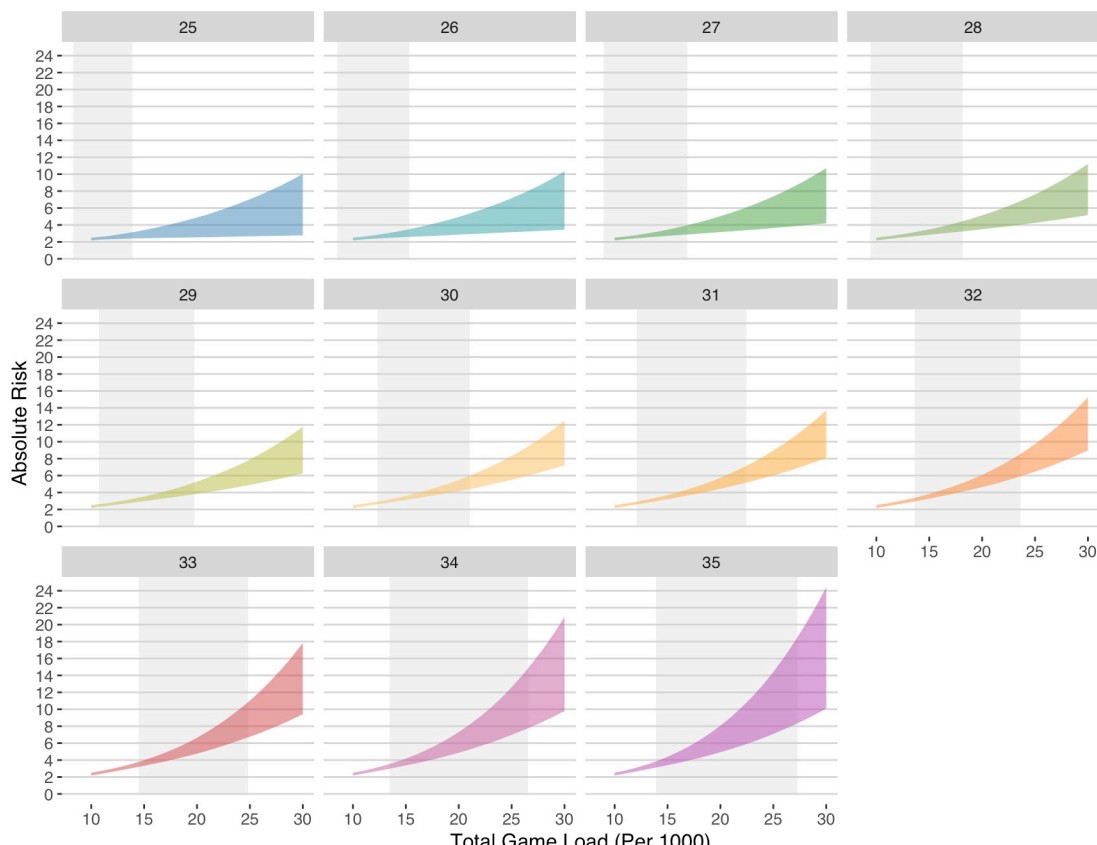

**Fig 7. Absolute risk time-loss 90% confidence bands associated with changing load and the modification effect of age according to the SNMM.** The shaded grey regions show the empirical 90th percentile range of actual game load observed for each age group.

analysis of tens of thousands of competition weeks over the complete professional careers of top male tennis players found significant increases in the risk of time-loss from competition with greater total competition load. We also demonstrated that the risk for the same increase in load increased with a player's biological age, indicating that the harmful effects of load are magnified for older players compared to younger players.

One of the strengths of the present study was its use of a structural nested mean model to identify the potential causal role of load in the presence of moderation by age. The SNMM is a recently developed technique that provides a more principled way to address observational study biases compared to standard regression techniques, the mainstay of epidemiological studies of injury in sport [32]. The SNMM is particularly necessary when the exposure and moderators of interest are varying in time, as is likely the case for any study of load and injury, owing to the dynamic nature of load and other putative factors involved in the mechanisms of injury. The main reason for this is that past exposure could influence levels of future exposure, moderators, and confounders. Careful balancing and conditioning at each time point of the analysis is required to remove the bias induced by the interplay of these factors over time [16].

Indeed, the present study found that standard methods, which do not protect against such biases, would lead to the conclusion that cumulative competition load was protective against absences from competition. One contributor to this discrepancy is age, which both theory and prior empirical evidence suggests is likely to modify the effect of load [33]. When player age is

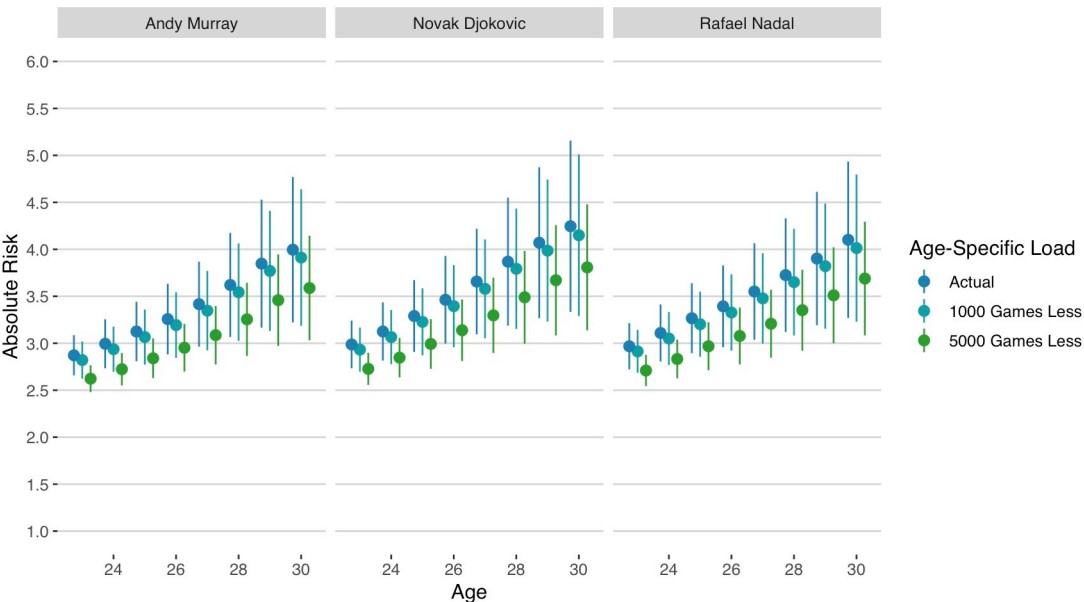

**Fig 8.  Absolute risk (± SD) of time-loss for actual age-specific total games played and counterfactual reductions of 1,000 and 5,000 games for Andy Murray, Rafael Nadal and Novak Djokovic.**

adjusted for with standard regression, it cuts off the effect of load because it is a correlate of future levels of load and a moderator of load's effect. Another contributor is the confounding of player rating, as greater ability predicts higher levels of future load but is also an independent predictor of a lower risk of time-loss. The counterintuitive protective effect of load has been frequently reported in studies of load and injury in sport [6]. The fact that we, in the context of time-loss in tennis, observe a reversal in effect when using causal inference methods raises the possibility that conclusions about the protective effects of load have been grounded in questionable methodology that has not adequately addressed effect modifiers or confounders. A recent critique of methods behind the literature on acute-chronic load ratios in the management of load, reinforces the dangers of observational study biases when researching the effects of dynamic exposures in sport [34].

Today's top men's tennis players compete in an average of 25 events per year and play an average of 50 games per event. Given current playing schedules, a reduction of 1,000 games of load is approximately equal to skipping an entire season of competition. Since the present study found only modest reductions in risk with 1,000 fewer games played, practically meaningful reductions in risk would appear to require early-career implementation of long-term load management strategies. Although reductions in load could be aided by structural changes in the sport, via a reduced calendar or shortened match format, for example, the commercial drives to see the best players playing longer and more often would suggest that immediate reductions in load will rest on the ability of players to sacrifice play opportunities and adopt more selective schedules.

Although the present study provides some broad guidance for load management in tennis, many questions remain to be addressed in order to develop more practically useful, individualized strategies for specific players. How effects of competition load vary by gender, playing surface, upper-body versus lower-body, or player anthropometry are all relevant questions for future research. Because all of these could be regarded as types of stratification analysis, the SNMM framework we have presented could be readily applied to address these questions. It

would also be possible to explore the severity of time-loss by considering the number of days between competitive play as the primary outcome of analysis or to link outcomes directly to injuries, as more consistent injury documentation becomes available.

Other major remaining questions would be difficult to address without resolving the limitations of present data and methodology. Our approach has presumed that cumulative external load is the primary mechanism by which load effects risk of injury. While total absolute load may be a reasonable measure of the cumulative stress on tissue, other factors about how this load is distributed over time may be important for further elucidation of the effects of load. The variability in load over time and the level of acute load relative to long-term load have both been stressed as important factors to the risk of tissue damage in athletes [35, 36]. Study into the casual effects of varying short- and long-term temporal patterns of load will require extending current causal methods for time-varying exposures and effect modifications that allow for greater flexibility in the exposure mechanism.

In focusing on load information that could be consistently observed for a large group of players over time, we have used game load as a proxy for the intensity of biomechanical stress player's are exposed to in a given match. Owing to differences in player movement and playing strategy, games played could mask significant differences in the duration and severity of experienced biomechanical stress.

Of even greater concern than the coarseness of available measures of competition load is the scarcity of training load. Without information about training load, the present study's ability to identify the potential causal effects of competition load rests on the assumption that competition load is the primary causative factor of time-loss and that, after accounting for observed covariates, the level of competition load is ignorable regardless of past training load. These are both strong assumptions that require further research to verify. Although the incompleteness of training load data is unlikely to allow for large-scale study, smaller scale investigations may be possible and still informative. Indeed, for epidemiological work of tennis injuries to have a meaningful impact on clinical practice, combining more principled statistical methods with a more complete picture of player load in training and competition will be a crucial next step.

## Conclusions

This study provides valuable new evidence about the potential causal role of cumulative competition load and time-loss events in professional tennis. Both the use of causal methods that are appropriate for dynamic dose-response mechanisms and the application of these methods to complete competition histories of hundreds of players makes this study's evidence for the harmful effects of load some of the strongest yet reported. We hope that our findings will highlight the need for casual methods in observational studies of injury in sport and will spark continued development and application of these techniques to further understand the causative role of load and guide future load management strategies.

## Supporting information

**S1 File. A document with a discussion of the incentives for competing in professional tennis, more detail on the definitions of a regular schedule and time-loss from competition, and some validatory analysis of the time-loss definition.**
(PDF)

## Author Contributions

**Conceptualization:** Stephanie A. Kovalchik.

**Data curation:** Stephanie A. Kovalchik.

**Formal analysis:** Stephanie A. Kovalchik.

**Investigation:** Stephanie A. Kovalchik.

**Methodology:** Stephanie A. Kovalchik.

**Project administration:** Stephanie A. Kovalchik.

**Software:** Stephanie A. Kovalchik.

**Supervision:** Stephanie A. Kovalchik.

**Validation:** Stephanie A. Kovalchik.

**Visualization:** Stephanie A. Kovalchik.

**Writing – original draft:** Stephanie A. Kovalchik.

**Writing – review & editing:** Stephanie A. Kovalchik.

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
