## [Decision Letter · Decision Letter 0]

14 Jan 2020

PONE-D-19-16725

`In Search of Lost Time': Identifying the causative role of cumulative competition load and competition time-loss in professional tennis using a structural nested mean model

PLOS ONE

Dear Dr. Kovalchik,

Thank you for submitting your manuscript to PLOS ONE. After careful consideration, we feel that it has merit but does not fully meet PLOS ONE’s publication criteria as it currently stands. Therefore, we invite you to submit a revised version of the manuscript that addresses the points raised during the review process.

Along with the revised version, we ask that you provide point-to-point responses to each of the comments raised by the reviewers. Please explain what revisions have been made within the manuscript to address each of the points raised. If a comment is not addressed, please justify accordingly in your response to the reviewer's comment.

In addition to the reviewers' comments, I would like to ask that any claims of causation are relaxed. You can state that you are studying the potential causes of some injury, but be careful not to state that your work identifies causes of injury. Further, Fig 4 presents a DAG and states that it is a causal model. The correct way to express this is to say that the DAG models the variables of interest under the assumption that the arcs represent causal influence; this is an important distinction (even though the DAG appears to follow some sort of a termporal order of events, rather than causation). Though I`m unsure why you would like to state this; a DAG under causal assumptions is generally used for simulating interventions, something which is not present in this study and hence, no point in making this assumption. Moreover, Figures 2 and 3 state that they present a DAG, but those graphs are not DAGs (i.e., nodes M and Age are parents of which node?).  

We would appreciate receiving your revised manuscript by Feb 28 2020 11:59PM. To enhance the reproducibility of your results, we recommend that if applicable you deposit your laboratory protocols in protocols.io, where a protocol can be assigned its own identifier (DOI) such that it can be cited independently in the future. For instructions see: http://journals.plos.org/plosone/s/submission-guidelines#loc-laboratory-protocols

We look forward to receiving your revised manuscript.

Kind regards,

Anthony C Constantinou

Academic Editor

PLOS ONE

Journal Requirements:

"The authors has declared that no competing interests exist."

We note that one or more of the authors are employed by a commercial company: Game Insight Group, Tennis Australia.

4. Please ensure that you refer to Figures 2-4 in your text as, if accepted, production will need this reference to link the reader to the figure.

Reviewers' comments:

Reviewer's Responses to Questions

**Comments to the Author**

1. Is the manuscript technically sound, and do the data support the conclusions?

Reviewer #1: Yes

Reviewer #2: Yes

2. Has the statistical analysis been performed appropriately and rigorously? 

Reviewer #1: Yes

Reviewer #2: Yes

3. Have the authors made all data underlying the findings in their manuscript fully available?

Reviewer #1: Yes

Reviewer #2: Yes

4. Is the manuscript presented in an intelligible fashion and written in standard English?

Reviewer #1: Yes

Reviewer #2: Yes

5. Review Comments to the Author

Reviewer #1: Specific questions, suggestions, and comments referenced to manuscript line number(s):

3: How exactly does injury threaten the sustainability of the sports industry?

12: The possessive pronoun “its” does not contain an apostrophe.

18: The pronoun “this” apparently refers to the “congested season” mentioned in the previous sentence. If this is the case, the “congested season” may “impose” stresses (the grind) on the best players, but the “congested season” would not “incur” them.

22: The phrase “the injury mechanism” implies that there is only one mechanism.

25: A better explanation of the “load” concept is needed. Most experts in the area would agree that “mechanical load” represents a combination of force (mass X acceleration) imposed on body tissues and the frequency of exposure within a specific time period. Others may define “load” in terms of physiological demand on the cardiorespiratory system in relation to some measure of exposure duration and/or frequency. The term “player load” is often used to refer to measurements derived from wearable technology that quantifies instantaneous changes in whole-body inertia over a defined amount of time.

31: The term “trainers” lacks specificity. Are you referring to coaches who guide strengthening and conditioning activities or “athletic trainers” who are charged with injury prevention and treatment?

66: Define “treatment weights” and explain how they address selection and confounding biases. What is meant by “treatment” and how are “weights” applied.

79-80: What is an “Elo-based” rating system? No reference is cited.

92-93: No information has been provided for the reader to have any understanding of the 1900 to 3000 range of points. Does rating groups of 100 points mean that there were 11 groups?

93-95: Does “a player rating of 2300 or higher” mean that the number of groups was reduced from 11 to 6? Explain why less than or equal to 3 weeks was chosen as a standard. The reference to “absences by month” (line 93) and the phrase “for at least 9 months of the year” (line 95) makes this content extremely hard to understand. Please be more explicit in explaining the basis for your operational definition of time loss.

98: The word “criterion” refers to a single standard. The word “criteria” is plural.

104: Please explain what is meant by “maximum gap” during a calendar month. Is this an alternate term for “absence from competition” mentioned in the first line sentence of the paragraph?

104-110: Does the reference to “linear mixed model” mean that you used a linear regression equation to estimate absence from competition for each player on a monthly basis? Line 108 refers to “a gap that was 2 weeks longer than expected,” but the Fig 1 legend refers to “maximum gap (in days)” between competitive events. This content does not clearly convey your definition of time loss from competition.

114-115: How can “gaps” defined as “25 to 40 days or fewer” represent a threshold for months that have only 30 or 31 days each? Does this mean the number of days of absence prior to the first competition during a given month? Surely, there is a way to explain your procedure in a manner that is more clearly understandable. A clearer distinction needs to be made between “elected” absence from competition and “unintended” absence attributed to injury. Figure 1 needs better explanation: Does 90th Percentile Range mean the range of 90th Percentile values for days of absence among the 389 players?

142: The referenced Fig is not designated by number (Fig 2?).

147: The referenced Fig is not designated by number (Fig 3?).

147-149: This clarification of the meaning of “load” should appear earlier in the manuscript (see the previous comment referenced to line 25).

158: The abbreviation “DAG” (directed acyclic graph?) is not defined in the text.

160: The referenced Fig is not designated by number (Fig 4?).

176: The abbreviation “MSM” (marginal structural model?) is not defined in the text.

180-182: The meaning of the phrase “previous levels of treatment” is not clear (see the previous comment referenced to line 66). Most readers are likely to interpret the word “treatment” as having something to do with therapeutic interventions following an injury.

185-216: I certainly do not possess the requisite level of knowledge about advanced mathematical modeling to appraise the quality of the content pertaining to the SNMM method.

297-309: I can follow the reasoning for the model specification, but I remain confused about the meaning of the term “treatment” in this section of the text.

330-332: This portion of the text refers to “competition week” (as well as the Fig 5 legend), but “Competition Age” is the label on the x-axis of the Fig. The latter term has not been introduced anywhere in the manuscript text. The y-axis “log(w)” label apparently refers to the log of inverse-probability of combined treatment weights and censor weights (lines 227-228 and the Fig 5 legend). Inconsistency in the use of terms and labels further confuses reader interpretation of the graph’s meaning.

333-335: Again, the term “competition weeks” appears in the text, but “Competition Age” is the label on the x-axis of the Fig.

353-354: The content in lines 350-351 connects the term “doubly-robust analysis” with the hazard ratio reported in the “SNMM Adjusted” column of Table 3 (5% increased in risk; 1.05). The “doubly robust estimates” of increased risk of time-loss for ages 25, 27, and 29 for an increase in game load of 1000 or more reported in line 354 apparently do not have corresponding hazard ratio values in Table 3, which complicates the reader’s understanding of the correspondence between information presented in the text with that presented in the Table.

401-409: This portion of the text provides the clearest explanation of the connection between the risk modeling methods and its results. After reading it, I finally figured out that “treatment weights” related to “player ability” and “past injury” as time-varying covariates. I strongly recommend making this connection much more explicit throughout the manuscript.

409-412: I suggest that content be added to the end of this sentence: “…questionable methodology that has not adequately addressed effect modifiers or confounders.”

Reviewer #2: Is the manuscript technically sound, and do the data support the conclusions?

Yes, but again – some of the statistics are beyond my understanding so very key to have a biostatistician review the paper to ensure this aspect.

PLOS Data policy:

I believe they have used only commercially available data from the tennis world – they have quoted the site – they have used several player names, again with only public data and also mention of an injury that is very public so it does not appear that any GDPR or HIPAA violations on the use of their data would be applicable.

Line 78 – please provide greater information for the reader on ‘player ratings’ – this can easily be confused with straight player rankings – ie number 1, 2, 4, in the world etc….. the ratings are important to the paper and many will not understand how this is calculated and how it is applicable. Would add this early in the manuscript.

Line 100 – Except for Grand Slams (4) and Indian Wells and Miami – which are essentially 10 day events and often could appear to have a gap in player competition days with early loss in IW, followed by no events to compete in until the next tournament. Also players with lower rankings often have very few competition opportunities during the month of March / if their ranking does not allow access to IW or Miami…..

Line 106 – so no direct injury illness reports were accessed, just player competition data….. you did a good job later in the paper stating this could be a limitation and that access to the player injury data could provide additional insight beyond what you have reported…. This is very good and true.

Line 110 – you mention a small sample to test this – was it like 8 players, or 90 players ? would be good to list the number so the reader knows – will add credibility to the data sample here.

Line 149 – Games…. Excellent – several prior epidemiological studies have found number of games to more closely represent player volume / load etc, compared to sets or matches which can be very misleading. Just as an aside, did you also look at points played ? or any other volume metrics, this would add additional information to the paper if you did study this but did not report it.

Line 152 – great point – Kibler et al, 1996 showed decreases in shoulder IR and total rotation ROM based on years of player and numbers of tournaments. This would parallel the statements you find and are reporting here that there is an effect of cumulative loading and age and that this does ultimately affect injury risk and time loss.

Line 274 - consider rewording sentence here ?

Line 418 – good point about limiting games, but likely as you state, 1000 or 5000 not practical due to exposure needed for ranking and success in the sport.

Line 454 – the the ?

Lines 450 – 460 – good discussion. For sure you bring up that training load in this study only represents competition load. There is limited ability to measure training load…. Which in many ways can be more repetitive and lead to injury away from competition with year round play inherent in the sport. With the advent of wearable technology, it may become more common to measure this parameter for researchers in the future, but at this time unlike other team sports with dedicated and consistent medical teams who measure this (training or practice) we may not have this aspect known in tennis for some time.

As a general rule, if you can increase the clinical application aspect of the paper, it would strengthen it for many readers of the journal. Several take a way points, what are the bottom lines from your amazing work ?

6. PLOS authors have the option to publish the peer review history of their article (what does this mean?). If published, this will include your full peer review and any attached files.

Reviewer #1: Yes: Gary B. Wilkerson

Reviewer #2: Yes: Todd Ellenbecker

---

## [Author Response · Author response to Decision Letter 0]

27 Jan 2020

PONE-D-19-16725

`In Search of Lost Time': Identifying the causative role of cumulative competition load and competition time-loss in professional tennis using a structural nested mean model’

I am thankful to the Editor and the two Referees for their constructive feedback. I am pleased that there was general agreement about the merits of the work and suggestions on areas that the manuscript could be improved. I am thankful for the opportunity to respond to those suggestions in a revision. 

Below is a point-by-point response to all of the feedback that was received. I believe the changes made in response to these suggestions have strengthened the paper, and I hope resolved any remaining concerns. Please note that in what follows the original comments are proceeded with ‘COMMENT’.

Editor Comments

COMMENT. In addition to the reviewers' comments, I would like to ask that any claims of causation are relaxed. You can state that you are studying the potential causes of some injury, but be careful not to state that your work identifies causes of injury.]

RESPONSE. The Editor makes an excellent point that any study based on observational evidence can attempt to estimate potential causes of outcomes. I’ve accordingly reviewed all of the instances were ‘causal’ claims are made in the paper and, where appropriate, replaced with ‘potential causes’. 

COMMENT. Further, Fig 4 presents a DAG and states that it is a causal model. The correct way to express this is to say that the DAG models the variables of interest under the assumption that the arcs represent causal influence; this is an important distinction (even though the DAG appears to follow some sort of a termporal order of events, rather than causation). 

RESPONSE. The Editor is correct that more precision was needed in the terminology around the DAG. I’ve added text in the description of Figure 4 in the main text, “...assuming that all edges capture the factors that have a causal influence on Y” and in the Figure caption as well (using ‘presumed causal model’).

COMMENT. Though I`m unsure why you would like to state this; a DAG under causal assumptions is generally used for simulating interventions, something which is not present in this study and hence, no point in making this assumption. Moreover, Figures 2 and 3 state that they present a DAG, but those graphs are not DAGs (i.e., nodes M and Age are parents of which node?). 

RESPONSE. In this case, the arrow between ‘M’ and the directed edge between X and Y in Figure 2 is denoting modification on the causal process between X and Y. This is distinct from direct or indirect modification. I believe this is an established way for a DAG to denote this type of effect modification (see for example, Weinberg, C. R. (2007). Can DAGs clarify effect modification?. Epidemiology (Cambridge, Mass.), 18(5), 569.). But I recognize that it may not be familiar to all readers so I have added text to explain what the “edge-to-edge” represents. 

Reviewer 1 Comments:

COMMENT. L3: How exactly does injury threaten the sustainability of the sports industry?

RESPONSE. After considering the Reviewer’s question, it was clear that ‘sustainability’ was not the appropriate word choice. I have revised the sentence to instead point to the potential economic costs to the sports industry when top athletes are unable to play due to injury with this rephrasing: ‘Injury is one of the most significant threats to the longevity of elite athletes and, when injury ends the careers of the industry's stars prematurely, can pose a significant threat to the business of sports’ (L2-3).

COMMENT. L12: The possessive pronoun “its” does not contain an apostrophe.

RESPONSE. The Reviewer is correct that there were three instances in the paper were the possessive pronoun incorrectly included an apostrophe. These have all been corrected in the revision.

COMMENT. L18. The pronoun “this” apparently refers to the “congested season” mentioned in the previous sentence. If this is the case, the “congested season” may “impose” stresses (the grind) on the best players, but the “congested season” would not “incur” them.

RESPONSE. I thank the Reviewer for pointing out the need for rephrasing here. I have replace ‘this incurs’ with ‘season schedule imposes’ to be more clear.

COMMENT. L22. The phrase “the injury mechanism” implies that there is only one mechanism.

RESPONSE. I agree that this implies a single mechanism, which oversimplifies the causes of injury. I’ve replaced all instances of ‘injury mechanism’ with ‘mechanisms of injury’.

COMMENT. L25: A better explanation of the “load” concept is needed. Most experts in the area would agree that “mechanical load” represents a combination of force (mass X acceleration) imposed on body tissues and the frequency of exposure within a specific time period. Others may define “load” in terms of physiological demand on the cardiorespiratory system in relation to some measure of exposure duration and/or frequency. The term “player load” is often used to refer to measurements derived from wearable technology that quantifies instantaneous changes in whole-body inertia over a defined amount of time.

RESPONSE. This is an excellent point. The wording in the paper implied a single definition of load which is far from the case. I’ve now added the following text to clarify that there are many definitions of load, which is one of the challenges to this area of research:

‘Gathering high-quality data about load is also a challenge. One reason for this is the lack of agreement on how `load` is defined. Load can take different meanings depending on the experts who are using it [10]: biomechanists use load to focus on the frequency and force of stress to joints, physiologists use load to refer to the respiratory demands on the cardiovascular system, while sports scientists use load to refer to total accelerations performed. Under any definition, a complete picture of the load an athlete may experience over time is rarely available owing to the difficulties of collecting data during the training periods of top athletes [11,12].’ (L34-42).

COMMENT. L31. The term “trainers” lacks specificity. Are you referring to coaches who guide strengthening and conditioning activities or “athletic trainers” who are charged with injury prevention and treatment?

RESPONSE. ‘Athletic trainers’ was the group referred to and this is what is used in the revision.

COMMENT. L66: Define “treatment weights” and explain how they address selection and confounding biases. What is meant by “treatment” and how are “weights” applied.

RESPONSE. The term “inverse probability of treatment weights” is a well-established technical term in causal inference. But I understand that some definition in the paper would help make this more approachable for a more general audience. I have added this in the Methods section with the lines: ‘...Imbalance in these factors are instead handled through the use of inverse probability of treatment weights. The `treatment', a general term the causal inference literature uses to refer to the main explanatory variable of interest, in this case is the cumulative competition load. Weighting observations by the inverse probability of the observed dose of treatment received is well known to be a more effective strategy for protecting against confounder bias than regression’. (L237-243)

COMMENT. L79-80: What is an “Elo-based” rating system? No reference is cited.

RESPONSE. I am grateful to the Reviewer for pointing out this oversight. This paper was still in press at the time of review. I’ve now added the paper’s citation as well as a reference to Arpad Elo’s original text on the system. I’ve also noted that the system is a ‘statistical algorithm for rating the latent ability of tennis players’ for further clarification of the basic goal of the system and how it differs from official rankings. 

COMMENT. L92-93: No information has been provided for the reader to have any understanding of the 1900 to 3000 range of points. Does rating groups of 100 points mean that there were 11 groups?

RESPONSE. The Reviewer is correct, there were 11 groups used in this descriptive analysis. I have also added a footnote to explain that a 1900 to 3000 would be a range in ratings that would capture players who are competing in Grand Slams, and what many in the sport would consider a minimal criteria for a ‘top player’.

COMMENT. L93-95: Does “a player rating of 2300 or higher” mean that the number of groups was reduced from 11 to 6? Explain why less than or equal to 3 weeks was chosen as a standard. The reference to “absences by month” (line 93) and the phrase “for at least 9 months of the year” (line 95) makes this content extremely hard to understand. Please be more explicit in explaining the basis for your operational definition of time loss.

RESPONSE. The Reviewer is correct that 2300 was a threshold for selecting the sample, though the players were not grouped any further in the actual study analysis. Those groups were only used in the exploration of a time-loss definition as it was anticipated that tournament entry would vary over the range of the ratings distribution. 

We agree that more justification for the time-loss definition was needed. The following text was added to the description of the reasons for the rule we applied:

‘Given that official rankings are based on a player's best 18 tournament results and that no ATP event outside of the World Tour Finals takes place in the months of November and December, it is reasonable to use 3 weeks as a threshold for the upper bound of between-event gaps of a typical top player’. (L101-105)

COMMENT. L98: The word “criterion” refers to a single standard. The word “criteria” is plural.

RESPONSE. I thank the Reviewer for catching this. The use of ‘criteria’ has been replaced with ‘criterion’.

COMMENT. L104: Please explain what is meant by “maximum gap” during a calendar month. Is this an alternate term for “absence from competition” mentioned in the first line sentence of the paragraph?

RESPONSE. For further clarification, I’ve added the following parenthetical statement after the introduction of ‘maximum gap’: ‘the largest number of consecutive days a player was not competing in a given month’. (L117)

COMMENT. L104-110: Does the reference to “linear mixed model” mean that you used a linear regression equation to estimate absence from competition for each player on a monthly basis? Line 108 refers to “a gap that was 2 weeks longer than expected,” but the Fig 1 legend refers to “maximum gap (in days)” between competitive events. This content does not clearly convey your definition of time loss from competition.

RESPONSE. The Reviewer is correct that a regression model was used as part of the definition of a time-loss event. The model was used to estimate what we would expect the max consecutive days between events (“gap days”) to be when a player is healthy and playing regularly. We then look for instances when the number of gap days in a month is 2 weeks longer or more than what the model would suggest is normal. I hope this is made more clear by the addition of the following explanation in the ‘Outcome’ section (L116-133):

“Definitions for time-loss from competition were individualized to each player using a linear mixed model of the maximum gap during a calendar month (the largest number of consecutive days a player was not competing in a given month) with player random effects. From this regression model, we could obtain the expected value for the maximum number of consecutive days a player spends away from competition in a given month. The model was trained on data for players who had 3 or more seasons at a rating of 2300 or more. For players with fewer than 3 seasons at the minimum rating, the expected maximum gap days was set to the average.

The above regression model provides a player and month specific estimate of the maximum between-event days (here on called `gap days') under normal conditions. A time-loss event was defined as instances where the actual gap days in a month were 2 weeks longer than expected (4 weeks longer for the month of January, owing to the off-season).”

For the Figure 1 caption, it now reads: “The 90th percentile range for ‘gap days’ for each month, which depicts the range containing 90% of the longest periods outside of competition that were observed for each month.”

COMMENT. L114-115: How can “gaps” defined as “25 to 40 days or fewer” represent a threshold for months that have only 30 or 31 days each? Does this mean the number of days of absence prior to the first competition during a given month? Surely, there is a way to explain your procedure in a manner that is more clearly understandable. 

RESPONSE. There did need to be an explanation of how we handled cases where the time between competition extended beyond one month. This is now addressed with the following addition to the revision: “Since a period between competition could extend over one or more months, the period was assigned to the month when the gap commenced and that month alone.” (L119-121)

COMMENT. A clearer distinction needs to be made between “elected” absence from competition and “unintended” absence attributed to injury. 

RESPONSE. The Reviewer makes an important point. We cannot know a player’s true intentions, so we have removed the word “unintended” and instead defined time-loss as an “extended break suggestive of an unintended absence” (L64-65).

COMMENT. Figure 1 needs better explanation: Does 90th Percentile Range mean the range of 90th Percentile values for days of absence among the 389 players?

RESPONSE. The Reviewer’s interpretation is exactly right. To make this clearer, the caption of Figure 1 has been rephrased as follows: “The 90th percentile range for `gap days' for each month, which depicts the range containing 90% of the longest periods outside of competition that were observed for each month.”

COMMENT. L142: The referenced Fig is not designated by number (Fig 2?). 

RESPONSE. The reference was an issue with the placement of the figure labels in LaTeX. I apologize for not having noticed this before the original submission. This has been corrected in the revision.

COMMENT. L147: The referenced Fig is not designated by number (Fig 3?).

RESPONSE. See comment on Fig 2 reference above.

COMMENT. L147-149: This clarification of the meaning of “load” should appear earlier in the manuscript (see the previous comment referenced to line 25).

RESPONSE. I completely agree. Please see my response to the comment on line 25.

COMMENT. L158: The abbreviation “DAG” (directed acyclic graph?) is not defined in the text.

RESPONSE. The abbreviation is now introduced in the first line of section ‘Exposure and moderator’.

COMMENT. L160: The referenced Fig is not designated by number (Fig 4?).

RESPONSE. See comment on Fig 2 reference above.

COMMENT. L176: The abbreviation “MSM” (marginal structural model?) is not defined in the text.

RESPONSE. The abbreviation ‘MSM’ is now introduced in the last paragraph of the Introduction.

COMMENT. L180-182: The meaning of the phrase “previous levels of treatment” is not clear (see the previous comment referenced to line 66). Most readers are likely to interpret the word “treatment” as having something to do with therapeutic interventions following an injury.

RESPONSE. “Treatment” is used as a general reference to the explanatory variable of interest in the causal inference literature. I know this might look strange to readers unfamiliar with this researchers but, on the other hand, it would be completely normal and clear for researchers who are. To try to strike a balance, I’ve added an explanation of this particular use of “treatment” in causal modelling at lines 239-241.

COMMENT. L297-309: I can follow the reasoning for the model specification, but I remain confused about the meaning of the term “treatment” in this section of the text.

RESPONSE. Please see the response to the previous comment.

COMMENT. L330-332: This portion of the text refers to “competition week” (as well as the Fig 5 legend), but “Competition Age” is the label on the x-axis of the Fig. The latter term has not been introduced anywhere in the manuscript text. 

RESPONSE. I agree with the Reviewer that it is important to have consistency in terminology. The revision has now replaced ‘competition age’ in Figures 5 and 6 with ‘competition weeks’.

COMMENT. The y-axis “log(w)” label apparently refers to the log of inverse-probability of combined treatment weights and censor weights (lines 227-228 and the Fig 5 legend). Inconsistency in the use of terms and labels further confuses reader interpretation of the graph’s meaning.

RESPONSE. The y-axis title in Figure 5 has been changed to “log(combined weight)” to be consistent with the text and caption.

COMMENT. L333-335: Again, the term “competition weeks” appears in the text, but “Competition Age” is the label on the x-axis of the Fig.

RESPONSE. Please see the comment in response to the note on 330-332.

COMMENT. L353-354: The content in lines 350-351 connects the term “doubly-robust analysis” with the hazard ratio reported in the “SNMM Adjusted” column of Table 3 (5% increased in risk; 1.05). The “doubly robust estimates” of increased risk of time-loss for ages 25, 27, and 29 for an increase in game load of 1000 or more reported in line 354 apparently do not have corresponding hazard ratio values in Table 3, which complicates the reader’s understanding of the correspondence between information presented in the text with that presented in the Table.

RESPONSE. I agree with the Reviewer that the description needed to be clearer here, especially in relating the summary in the text to the numbers in Table 3. The following edits have been made to that description in order to improve its clarity: “Comparisons between the lowest 25th and highest 25th percentiles of empirical load observed at ages 25, 27 and 29 showed even starker effects. Based on the doubly-robust

estimates, the hazard ratios of players in the top 25% of load were consistently greater than those of players of the same age but with the lowest 25% of experienced load. At age 25, the hazard ratio of 1.17 (95% CI 1.02-1.35) corresponds to a risk increase of 17%; at age 27, the hazard ratios of 1.28 (95% CI 1.12-1.47) and 1.06 (95% CI 1.03-1.10) correspond to an increase of 21%; and at age 29, the hazard ratios of 1.65 (95% CI 1.38-1.95) and 1.33 (95% CI 1.20-1.47) correspond to an increase of 24%. Though each of these comparisons correspond to a fixed increase of 3,000 games, we see that the risk associated with that same change in load is increasing, indicating the positive effect modification due to age.” (L381-391)

COMMENT. L401-409: This portion of the text provides the clearest explanation of the connection between the risk modeling methods and its results. After reading it, I finally figured out that “treatment weights” related to “player ability” and “past injury” as time-varying covariates. I strongly recommend making this connection much more explicit throughout the manuscript.

RESPONSE. To help link the ‘treatment weights’ to these covariates, we’ve revised the following sentence that appears early in the ‘Estimation’ section: “The purpose of these

weights is to create a pseudo population that is balanced with respect

to confounding variables, like player ability or competitive play, for all time t.” (L256-258)

COMMENT. L409-412: I suggest that content be added to the end of this sentence: “…questionable methodology that has not adequately addressed effect modifiers or confounders.”

RESPONSE. Thank you for this suggestion. The additional text has been added to this statement in the revised ‘Discussion’.

Reviewer 2 Comments:

COMMENT. Line 78 – please provide greater information for the reader on ‘player ratings’ – this can easily be confused with straight player rankings – ie number 1, 2, 4, in the world etc….. the ratings are important to the paper and many will not understand how this is calculated and how it is applicable. Would add this early in the manuscript.

RESPONSE. The Reviewer is correct that a better explanation of the player ratings was needed. The following has been added to this section to explain that the ratings are a ‘statistical algorithm for rating the latent ability of tennis players’. In addition, two references have been added that describe the Elo system and the specific version we are using in the paper. I hope this clarifies the basic goal of the system and how it differs from official rankings. 

COMMENT. Line 100 – Except for Grand Slams (4) and Indian Wells and Miami – which are essentially 10 day events and often could appear to have a gap in player competition days with early loss in IW, followed by no events to compete in until the next tournament. Also players with lower rankings often have very few competition opportunities during the month of March / if their ranking does not allow access to IW or Miami…..

RESPONSE. These are all excellent points. In fact, we do observe some of the top players who have some of their longest breaks (in the absence of injury) in February. These player- and month-specific effects are exactly why we needed to use a model approach with these exact variables in order to identify potential time-loss events for any individual player, as, for example, a 4-week gap going into March might be entirely normal for some players but unusual for others.

COMMENT. Line 106 – so no direct injury illness reports were accessed, just player competition data….. you did a good job later in the paper stating this could be a limitation and that access to the player injury data could provide additional insight beyond what you have reported…. This is very good and true.

RESPONSE. I am pleased that this crucial point came across. Yes, we have focused on when a player is and isn’t competing, which has the advantage that it can be completely observed for all professional players throughout their career. But the Reviewer rightly points out that, by not using direct injury data, we are at best only getting indirectly at a subset of injury events.

COMMENT. Line 110 – you mention a small sample to test this – was it like 8 players, or 90 players ? would be good to list the number so the reader knows – will add credibility to the data sample here.

RESPONSE. The Reviewer makes an excellent point. We have added the sample size of this validation, which was for three players, to the revision. Unfortunately, the injury history on public sources, like Wikipedia, are only well documented for a handful of players, otherwise studying injury in tennis would be much more straightforward!

COMMENT. Line 149 – Games…. Excellent – several prior epidemiological studies have found number of games to more closely represent player volume / load etc, compared to sets or matches which can be very misleading. Just as an aside, did you also look at points played ? or any other volume metrics, this would add additional information to the paper if you did study this but did not report it.

RESPONSE. This is a great question. While match statistics, like service points played or total points played, have quite a long history for Grand Slams, they are not as well documented for other events. Game and Set scores, however, have been recorded for all professional matches for decades. So we focused exclusively on total games. As statistical documentation for matches continues to improve over time, I suspect in a few years we may have at least a decade of more detailed match statistics for all professional matches and the question of these alternative measures of load could be investigated.

COMMENT. Line 152 – great point – Kibler et al, 1996 showed decreases in shoulder IR and total rotation ROM based on years of player and numbers of tournaments. This would parallel the statements you find and are reporting here that there is an effect of cumulative loading and age and that this does ultimately affect injury risk and time loss.

RESPONSE. I am grateful to the Reviewer for bringing this study to my attention. I have now added this reference in the Discussion as additional support for the effect modification of age. “One contributor to this discrepancy is age, which both theory and prior empirical evidence suggests is likely to modify the effect of load (Kibler et al. 1996)”.

COMMENT. Line 274 - consider rewording sentence here ?

RESPONSE. The Reviewer is correct that this sentence was not clearly worded. The following has been used in its place in the revision: “For all hazard ratios shown, the reference player was a 25 year-old with a total competition load of 10,000 games.”

COMMENT. Line 418 – good point about limiting games, but likely as you state, 1000 or 5000 not practical due to exposure needed for ranking and success in the sport.

RESPONSE. Yes, one implication of these findings is that, without the structures of tennis making an effort to change the current scheduling demands, the practical barriers to making meaningful reductions in competition load are considerable. I am glad that these comes across in the paper’s Discussion.

COMMENT. Line 454 – the the ?

RESPONSE. Thank you for pointing out this type. This has been corrected in the revision.

COMMENT. Lines 450 – 460 – good discussion. For sure you bring up that training load in this study only represents competition load. There is limited ability to measure training load…. Which in many ways can be more repetitive and lead to injury away from competition with year round play inherent in the sport. With the advent of wearable technology, it may become more common to measure this parameter for researchers in the future, but at this time unlike other team sports with dedicated and consistent medical teams who measure this (training or practice) we may not have this aspect known in tennis for some time.

As a general rule, if you can increase the clinical application aspect of the paper, it would strengthen it for many readers of the journal. Several take a way points, what are the bottom lines from your amazing work ?

RESPONSE. I agree with the Reviewer that the study would be strengthened if there was a more direct link to clinical applications. However, given the preliminary nature of this study, the first true causal inference approach to tennis competition loss, and the lack of training data in the measure of load, I would caution against drawing too many implications for clinical decision-making at this stage. But I do see this as an important step towards such work and I hope the paper better points to this with the addition of the following statement in the ‘Discussion’: “Indeed, for epidemiological work of tennis injuries to have a meaningful impact on clinical practice, combining more principled statistical methods with a more complete picture of player load in training and competition will be a crucial next step.” (L495-L598).

---

## [Decision Letter · Decision Letter 1]

18 Mar 2020

PONE-D-19-16725R1

`In Search of Lost Time': Identifying the causative role of cumulative competition load and competition time-loss in professional tennis using a structural nested mean model

PLOS ONE

Dear Dr. Kovalchik,

Thank you for submitting your manuscript to PLOS ONE. After careful consideration, we feel that it has merit but does not fully meet PLOS ONE’s publication criteria as it currently stands. Therefore, we invite you to submit a revised version of the manuscript that addresses the points raised during the review process.

As you will see, the reviewers are generally happy with the revised version. Reviewer #3 has identified a few more minor revisions, which need to be addressed before the manuscript is accepted for publication.

We would appreciate receiving your revised manuscript by May 02 2020 11:59PM. To enhance the reproducibility of your results, we recommend that if applicable you deposit your laboratory protocols in protocols.io, where a protocol can be assigned its own identifier (DOI) such that it can be cited independently in the future. For instructions see: http://journals.plos.org/plosone/s/submission-guidelines#loc-laboratory-protocols

We look forward to receiving your revised manuscript.

Kind regards,

Anthony C Constantinou

Academic Editor

PLOS ONE

Reviewers' comments:

Reviewer's Responses to Questions

**Comments to the Author**

1. If the authors have adequately addressed your comments raised in a previous round of review and you feel that this manuscript is now acceptable for publication, you may indicate that here to bypass the “Comments to the Author” section, enter your conflict of interest statement in the “Confidential to Editor” section, and submit your "Accept" recommendation.

Reviewer #1: All comments have been addressed

Reviewer #2: All comments have been addressed

Reviewer #3: (No Response)

2. Is the manuscript technically sound, and do the data support the conclusions?

Reviewer #1: Yes

Reviewer #2: Yes

Reviewer #3: Yes

3. Has the statistical analysis been performed appropriately and rigorously? 

Reviewer #1: Yes

Reviewer #2: Yes

Reviewer #3: Yes

4. Have the authors made all data underlying the findings in their manuscript fully available?

Reviewer #1: Yes

Reviewer #2: Yes

Reviewer #3: Yes

5. Is the manuscript presented in an intelligible fashion and written in standard English?

Reviewer #1: Yes

Reviewer #2: Yes

Reviewer #3: Yes

6. Review Comments to the Author

Reviewer #1: Responses to the specific review comments were very thorough and the related manuscript revisions have been done well.

Reviewer #2: Thank you for the opportunity to re-review this fine work. The authors have provided a thoughtful and effective response to all queries imposed after the initial review process in my opinion. I do not have any additional comments or requirements for this paper prior to acceptance for publication. I feel this paper will be an excellent addition to the literature in this area for tennis. The authors should be congratulated for a fine work as well as the other reviewer who was able to review and comment on very technical statistical aspects of the paper which provided exceptional rigor to this review process.

Reviewer #3: The study sought to identify potential causative factors associated with injury in male tennis players (n=389) by focusing on competition load and time loss to competition using a structural nested mean model. The total load significantly increased the risk of time-loss with a hazard rate of 1.05 per 1,000 games.

Minor revisions:

1- Line 117: State the covariance structure used in the linear mixed model and the criteria for selecting it.

2- Line 131: Indicate how the small sample of three players was identified.

3- State and justify the study’s target sample size with a pre-study statistical power calculation. The power calculation should include: sample size, alpha level (indicating one or two-sided), minimal detectable difference and statistical testing method.

4- Cite the statistical software used for the analysis.

7. PLOS authors have the option to publish the peer review history of their article (what does this mean?). If published, this will include your full peer review and any attached files.

Reviewer #1: Yes: Gary B. Wilkerson, EdD, ATC

Reviewer #2: Yes: Todd S. Ellenbecker, DPT, MS, SCS, OCS, CSCS

Reviewer #3: No

---

## [Author Response · Author response to Decision Letter 1]

20 Mar 2020

I want to thank the Referees for their latest set of comments. I recognize that there were some minor revisions that were still needed. Below I have detailed each suggestion and how I've responded to it in the latest version (responses given in []). I hope these have addressed all remaining concerns.

1- Line 117: State the covariance structure used in the linear mixed model and the criteria for selecting it.

[I’ve now added that an unstructured covariance-variance structure was used, which is the default for the lme4 package in R and has the advantage that it makes no assumptions about the within-player covariance-variance.]

2- Line 131: Indicate how the small sample of three players was identified.

[The sample was chosen by identifying former World No. 1 players who have been competitive within the past decade. Within this group, the Wikipedia pages were reviewed and the subset with detailed notes on their play activity in each season were selected for the validation sample. ]

3- State and justify the study’s target sample size with a pre-study statistical power calculation. 

The power calculation should include: sample size, alpha level (indicating one or two-sided), minimal detectable difference and statistical testing method.

[The goal of the study was to have a census of recent top players who we could consider as “regular” tour players. Thus, this was the primary goal in deciding on the exclusion criteria and therefore sample size for the study. However, we agree with the reviewer that even if power was not the primary driver for the study sample, it is still important to consider when interpreting the study results. If we were to use a traditional survival analysis to compare the 25% of players with the greatest game load to the rest of the sample, we estimate we would have 80% power to detect a 20% increase or more in the risk of competition time loss compared to a base rate of 3%. This analysis has been added to the data description in the revision (L114-116).]

4- Cite the statistical software used for the analysis.

[All the analysis was conducted in the R programming language, which I have indicated in the revision along with a citation for the language (L 323-324).]

---

## [Editor Report · Decision Letter 2]

27 Mar 2020

`In Search of Lost Time': Identifying the causative role of cumulative competition load and competition time-loss in professional tennis using a structural nested mean model

PONE-D-19-16725R2

Dear Dr. Kovalchik,

We are pleased to inform you that your manuscript has been judged scientifically suitable for publication and will be formally accepted for publication once it complies with all outstanding technical requirements.

With kind regards,

Anthony C Constantinou

Academic Editor

PLOS ONE
---

## [Editor Report · Acceptance letter]

31 Mar 2020

PONE-D-19-16725R2 

‘In Search of Lost Time’: Identifying the causative role of cumulative competition load and competition time-loss in professional tennis using a structural nested mean model 

Dear Dr. Kovalchik:

I am pleased to inform you that your manuscript has been deemed suitable for publication in PLOS ONE. Congratulations! Your manuscript is now with our production department. 

With kind regards,

on behalf of

Dr. Anthony C Constantinou 

Academic Editor

PLOS ONE